# Is an opportunistic primary care-based intervention for non-responders to bowel screening feasible and acceptable? A mixed-methods feasibility study in Scotland

Natalia Calanzani,[1] Debbie Cavers,[1] Gabriele Vojt,[2] Sheina Orbell,[3] Robert J C Steele,[4] Linda Brownlee,[5] Steve Smith,[6] Julietta Patnick,[7] David Weller,[1] Christine Campbell[1]

For numbered affiliations see end of article.

**Correspondence to**
Natalia Calanzani;
natalia.calanzani@ed.ac.uk

## ABSTRACT

**Objectives** We aimed to test whether a brief, opportunistic intervention in general practice was a feasible and acceptable way to engage with bowel screening non-responders.

**Design** This was a feasibility study testing an intervention which comprised a brief conversation during routine consultation, provision of a patient leaflet and instructions to request a replacement faecal occult blood test kit. A mixed-methods approach to evaluation was adopted. Data were collected from proformas completed after each intervention, from the Bowel Screening Centre database and from questionnaires. Semi-structured interviews were carried out. We used descriptive statistics, content and framework analysis to determine intervention feasibility and acceptability.

**Participants** Bowel screening non-responders (as defined by the Scottish Bowel Screening Centre) and primary care professionals working in five general practices in Lothian, Scotland.

**Primary and secondary outcome measures** Several predefined feasibility parameters were assessed, including numbers of patients engaging in conversation, requesting a replacement kit and returning it, and willingness of primary care professionals to deliver the intervention.

**Results** The intervention was offered to 258 patients in five general practices: 220 (87.0%) engaged with the intervention, 60 (23.3%) requested a new kit, 22 (8.5%) kits were completed and returned. Interviews and questionnaires suggest that the intervention was feasible, acceptable and consistent with an existing health prevention agenda. Reported challenges referred to work-related pressures, time constraints and practice priorities.

**Conclusions** This intervention was acceptable and resulted in a modest increase in non-responders participating in bowel screening, although outlined challenges may affect sustained implementation. The

## Strengths and limitations of this study

► This intervention is grounded in psychological theory and evidence on factors associated with non-participation in bowel screening.
► Furthermore, it considered the pragmatic reality of a dynamic, time-pressured primary care environment.
► This is a small-scale, non-randomised feasibility study in one region of Scotland, targeting non-responders who consult in primary care.
► As the intervention is of an opportunistic nature, data on patient characteristics (such as medical history or ethnicity) are not available.
► Further limitations include not being able to record information on non-responders and ascertain how many were missed; in addition to the low participation rate among general practices invited to take part in the study.

strategy is also aligned with the increasing role of primary care in promoting bowel screening.

## INTRODUCTION

Bowel screening using a faecal occult blood test (FOBt) enables identification of earlier stage cancers when treatment is more likely to be beneficial,[1] ultimately leading to reduction in bowel cancer mortality.[2] The UK has well-established bowel screening programmes in each of its constituent countries. The Scottish Bowel Screening Centre (SBSC) sends a guaiac-based FOBt biennially to eligible patients aged 50–74 years.[3] The current uptake is 57.7%; with lower participation among the most deprived populations compared with the least deprived groups (45.5% vs 66.6%, respectively). Uptake is higher for women (60.6%) compared with men (54.7%).[4]

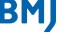

Both barriers to uptake and effective strategies to increase participation in bowel screening are described in the literature. Lack of awareness of bowel cancer[5] or of screening,[6] concerns about unpleasantness and embarrassment,[7 8] fear of the outcome,[5 9] fatalism[10] and perception of risk[11 12] are commonly identified barriers. Nevertheless, there is good evidence that reminders targeting patients[13 14] and physicians,[5] having one-on-one interactions/education with general practitioners (GPs) and/or nurses[6 15] and GP endorsements[6 16] have a positive impact on screening uptake.

Primary care has an important and increasing role in cancer prevention and cancer screening.[17] In the UK alone, a number of interventions involving primary care have been recently developed.[1 13 16 18 19] In Scotland, a government programme aiming to improve cancer survival (the Detect Cancer Early Programme)[20] provided a financial incentive for practices meeting defined bowel screening targets.[21 22]

In this context, we aimed to test the feasibility and acceptability of an opportunistic intervention in general practice patient consultations, examining whether a brief conversation was a viable way to engage with non-responders and increase bowel screening participation. The study was undertaken in the Lothian region of Scotland which has slightly lower bowel screening uptake (57.2%) than the national average, and shows similar variation based on sex and socioeconomic status.[4]

## MATERIALS AND METHODS
### Recruitment of practices and patients
#### Practice recruitment
National Health Service (NHS) Lothian provided the research team with a list of 112 practices in this region. The list had information on practice code, per cent screening uptake in 2013, practice list size, number of patients aged 50–75, number of average monthly non-responders, mean Scottish Index of Multiple Deprivation decile[23] (for those aged 50–75) and whether or not practices took part in the bowel Scottish Quality and Outcomes Framework (QOF).[21] Eleven general practices were purposively selected for a first wave of recruitment. We oversampled among the most deprived practices with lower uptake (as it was perceived that these practices could benefit the most from the intervention), while also taking into account the other factors listed above (as these would impact on how many patients could potentially be approached during the study period). Practices were invited to take part in the study via a personalised email sent by the study's principal investigator.

A visit was scheduled at the practices that were interested in taking part in the study. A brief information session was delivered, giving background information on colorectal cancer and screening, known barriers and facilitators to screening uptake and a thorough description of the study. Practices also received a folder containing the intervention materials, study information sheet, ethical approvals, background information on bowel cancer and bowel screening and a consent form.

#### Estimated size of study population
A preliminary calculation estimated that each recruited general practice would have the opportunity to engage with up to 182 potentially eligible patients during the study period (see online supplementary file 1). These figures are rough estimates as the study findings will guide calculations of a powered sample size for a larger study. We aimed to recruit up to six general practices for testing the intervention's feasibility and acceptability.

#### Patient inclusion and exclusion criteria
The target population were men and women aged 50–74 registered in a participating Lothian practice who received an invitation to screening and did not return a completed kit with a definitive screening result within 90 days (ie, the official SBSC's definition of a non-responder). Patients were excluded from the study when they lacked mental capacity as defined by the Mental Capacity Act 2005[24] and at professionals' discretion where patients were regarded as too ill (eg, undergoing cancer treatment or in receipt of palliative care) or distressed to take part.

#### Identifying bowel screening non-responders
NHS Lothian and the SBSC routinely provide Scottish general practices with a list of non-responders. The research team worked alongside each participating practice to create a customised plan to ensure they could efficiently flag non-responders in their computer systems if they did not already have a system in place.

### Intervention
The intervention comprised a brief conversation about bowel screening with non-responders. During a consultation with an eligible patient, the primary care professional (PCP) (a GP, practice nurse or healthcare assistant) raised the topic of non-participation using neutral statements and discussed any patient concerns. A leaflet with further information and an opportunity to request a bowel screening kit (via email, phone or tear off slip with Freepost) was offered. The intervention was designed to last 3–5 min. As part of the intervention, patients could also choose whether or not to develop a written plan of how to complete and return the kit (an implementation intention).[25] In addition to the information leaflet and Freepost envelope, the intervention was supported by: an A5 set of three to four suggested questions/topics for discussion; an intervention flowchart and guidance sheet for PCPs (see online supplementary file 1).

In developing the intervention content, we drew on our previous work on strategies promoting uptake of FOBt screening[26] and available literature on factors associated with uptake and barriers to screening. We also drew on psychological models, principally on Implementation Intentions,[25] the Health Behaviour Framework,[27] and were guided by principles of motivational interviewing[28] and informed choice.[29] We chose the Health Behaviour

Framework as it synthesises major health behaviour models while also considering contextual factors. It has also been successfully applied to cancer screening health behaviours.[30–32] Implementation Intentions have been associated with higher participation in studies about cervical,[33] breast[34] and bowel[35] screening. Finally, motivational interviewing has been shown to increase bowel screening uptake[36 37]; its principles aided the development of non-directive statements to discuss non-participation.

The process of developing the intervention also included eight interviews with health professionals and 19 non-responders to bowel screening to explore their views on its acceptability in a primary care setting. Overall feedback was positive and results are reported elsewhere (manuscript under review). We sought approaches (and wording in our materials) which were not coercive but invited participants to consider the offer of screening after balancing potential benefits and harms. Materials conformed to the Scottish Bowel Screening Programme[38] and the NHS Cancer Screening Programme[39] resources and guidance.

### Data collected for evaluation
#### Delivery of the intervention
PCPs logged details of each intervention on a proforma (see online supplementary file 1). Researchers regularly visited practices to collect these and to distribute materials as required, recording all communication/events in an intervention log. The intervention was planned to run for 3–4 months in each practice, depending on availability.

#### Requests for screening kits
Requests for new kits were made to the SBSC, which logged both the requests using the tear-off slip and the returned kits. It was not possible to identify email or telephone requests relating to this project due to the high volume received through these means daily. For the purposes of comparison, the SBSC also provided data on total number of requests for a replacement kit made by each of the recruited practices during the intervention period (plus one extra month to allow time for requests

to come in) and for equivalent periods at 6 months, 1 year and 2 years before the intervention.

#### Questionnaire and interview data
The feasibility parameters and mechanisms to be investigated are available in online supplementary file 1. Brief end of study questionnaires were developed for primary care and bowel screening staff. The questionnaires comprised closed and open-ended questions and focused on intervention acceptability and potential impact on workload. Semi-structured interviews were carried out with members of the practice team (aiming to interview the practice manager and at least one GP or practice nurse). Interviews sought to ascertain views on the running of the brief intervention, its acceptability and its overall feasibility as part of routine primary care.

#### Data analysis
Templates were created using SPSS V.19 for Windows[40] to collect data on the practice proforma and the end of study questionnaires. Quantitative data from the SBSC, proformas and end of study questionnaires were analysed using descriptive statistics (summaries, frequencies and cross-tabulations). As we had a purposeful sample of practices and a non-random sample of patients, no inferential statistics were calculated.[41]

End of study interviews were digitally recorded and transcribed verbatim. Qualitative data were analysed using thematic analysis informed by a framework approach including techniques of familiarisation, coding, indexing, charting, mapping and interpretation[42 43] assisted by QSR NVivo V.7 software.[44] This approach was considered appropriate due to the pre-existing feasibility parameters being tested. Identified themes were considered in the context of these parameters and interpreted according to existing theory and research. Scrutiny both within and across transcripts ensured that the analysis encompassed all perspectives and used the whole dataset. All transcripts were read by two researchers (DC and NC) and 50% were subject to triple initial coding (DC, NC and CC). The initial coding framework was developed by DC and was reiterated following discussion, with any discrepancies explored and accounted

**Table 1** Characteristics of recruited practices

| Recruited practices | Location | Uptake % (2013) | Pop 50–75 | Mean SIMD decile (50–75 year olds)* | Average monthly non-responders | Practice list size | Signed up to QOF | Start date | End date |
|---|---|---|---|---|---|---|---|---|---|
| Practice A | Edinburgh | <45% | 1413 | 2.6 | 41 | 6888 | No | 04/03/15 | 05/07/15 |
| Practice B | Edinburgh | 45%–50% | 2654 | 3.6 | 61 | 10440 | Yes | 14/04/15 | 15/08/15 |
| Practice C | East Lothian | 50%–55% | 2515 | 4.5 | 56 | 8693 | Yes | 22/04/15 | 03/09/15 |
| Practice D | Edinburgh | 50%–55% | 1241 | 6.2 | 29 | 5326 | Yes | 20/04/15 | 24/08/15 |
| Practice E | Midlothian | 55%–60% | 1668 | 5.5 | 30 | 5201 | Yes | 05/03/15 | 08/07/15 |

*The SIMD is a measure of multiple deprivation which combines different domains related to employment, income, health, education, skills and training, geographic access to services, crime and housing.[23] The lower the decile number, the higher the deprivation levels.
Pop, population; QOF, Quality and Outcomes Framework; SIMD, Scottish Index of Multiple Deprivation.

**Table 2** Patient and staff characteristics

| Overall data | Practice A n (%) | Practice B n (%) | Practice C n (%) | Practice D n (%) | Practice E n (%) | Total n (%) |
|---|---|---|---|---|---|---|
| Patient sex | | | | | | |
| Men | 43 (60.6) | 25 (45.5) | 8 (28.6) | 43 (61.4) | 18 (52.9) | 137 (53.1) |
| Women | 28 (39.4) | 30 (54.5) | 20 (71.4) | 27 (38.6) | 16 (47.1) | 121 (46.9) |
| Total | 71 (100.0) | 55 (100.0) | 28 (100.0) | 70 (100.0) | 34 (100.0) | 258 (100.0) |
| Patient age* | | | | | | |
| Median (IQR) | 55.50 (53.00–64.00) | 55.50 (51.75–62.25) | 58.50 (52.25–67.75) | 63.00 (56.25–69.00) | 59.00 (53.75–63.00) | 58.00 (53.00–65.00) |
| 50–54 | 25 (35.7) | 24 (44.4) | 10 (35.7) | 13 (19.1) | 11 (32.4) | 83 (32.7) |
| 55–59 | 22 (31.4) | 9 (16.7) | 5 (17.9) | 14 (20.6) | 7 (20.6) | 57 (22.4) |
| 60–64 | 8 (11.4) | 10 (18.5) | 4 (14.3) | 11 (16.2) | 11 (32.4) | 44 (17.3) |
| 65–69 | 6 (8.6) | 6 (11.1) | 4 (14.3) | 15 (22.1) | 5 (14.7) | 36 (14.2) |
| 70–74 | 9 (12.9) | 4 (7.4) | 3 (10.7) | 10 (14.7) | 0 (0.0) | 26 (10.2) |
| 75–79 | 0 (0.0) | 1 (1.9) | 2 (7.1) | 5 (7.4) | 0 (0.0) | 8 (3.1) |
| Total | 70 (100.0) | 54 (100.0) | 28 (100.0) | 68 (100.0) | 34 (100.0) | 254 (100.0) |
| Interventions by primary care role | | | | | | |
| GP | 33 (46.5) | 31 (56.4) | 21 (75.0) | 50 (71.4) | 31 (91.2) | 166 (64.3) |
| PN | 38 (53.5) | 11 (20.0) | 7 (25.0) | 20 (28.6) | 3 (8.8) | 79 (30.6) |
| HCA | 0 (0.0) | 13 (23.6) | 0 (0.0) | 0 (0.0) | 0 (0.0) | 13 (5.0) |
| Total | 71 (100.0) | 55 (100.0) | 28 (100.0) | 70 (100.0) | 34 (100.0) | 258 (100.0) |

Missing data: There were no missing data for patient sex and staff carrying out the intervention. There were four missing cases for patient age. Percentages may not add up to 100 due to rounding.

*Eight patients aged 75 or older were included as their last invitation to screening happened before their 75th birthday (hence meeting eligibility criteria).

GP, general practitioner; HCA, healthcare assistant.; PN, practice nurse.

for. Content analysis[45] was used to summarise, categorise and interpret text entries made by practice staff on proformas to record reasons for consultation.

### Ethical approval and consent

The study was approved by the South East Scotland Research Ethics Committee 01 (reference 14/SS/1067) and the NHS Lothian's Research and Development Office (Project Number 2014/0366). The Scottish Bowel Screening Governance Reference Group also approved the study. Written informed consent was obtained via the practice manager or GP partner and separate consent was obtained for end of study interviews. Practices were reimbursed for their participation.

## RESULTS

### Practice recruitment

Six out of 11 invited practices consented to participate in the study; one practice subsequently withdrew due to resource issues (online supplementary file 2). The five remaining practices varied in size, bowel screening uptake and deprivation levels (table 1). All but one practice were signed up to the Scottish QOF.

### Setting up the intervention

Although all practices were routinely provided with an electronic list of non-responders, there was variation in the methods in place to identify non-responders during consultations. In two practices, the researchers coded non-responders into practices' computer systems, also helping to insert screen 'pop-up' reminders in one of these cases. The remaining three practices already had systems in place.

One practice developed a digital proforma in their GP system instead of using the paper-based one provided by the research team. Planned monthly visits were not always required and were adapted to suit practice needs.

### Intervention delivery, acceptance and impact on screening uptake

Overall, 258 patients were approached between March and September 2015. Men were approached slightly more often than women (53.1% vs 46.9%) and most patients were among the younger eligible age groups for screening (median 58.00, IQR 53.00–65.00) (table 2). No information on patient ethnicity was available. The median duration of the intervention was 2.00 min (IQR 1.25–5.00).

**Table 3** Intervention acceptability, requested and returned kits

| Overall data | | Practice A n (%) | Practice B n (%) | Practice C n (%) | Practice D n (%) | Practice E n (%) | Total n (%) |
|---|---|---|---|---|---|---|---|
| Duration of intervention (minutes)* | Median (IQR) | 3.50 (2.00–5.00) | 2.00 (2.00–4.50) | 2.00 (1.00–5.00) | 2.00 (2.00–3.25) | 1.00 (1.00–2.00) | 2.00 (1.25–5.00) |
| Acceptance of intervention | Accepted (yes) | 56 (78.9) | 46 (85.2) | 24 (96.0) | 61 (88.4) | 33 (97.1) | 220 (87.0) |
| | Leaflet given (yes) | 40 (57.1) | 44 (81.5) | 24 (85.7) | 48 (68.6) | 34 (100.0) | 190 (74.2) |
| | Leaflet completed in practice (yes) | 10 (14.3) | 9 (16.7) | 0 (0.0) | 19 (27.9) | 0 (0.0) | 38 (17.3) |
| Requested kits using a slip | N requested kits/total interventions | 16/71 | 7/55 | 6/28 | 25/70 | 6/34 | 60/258 |
| | (% requesting a kit among total interventions) | (22.5) | (12.7) | (21.4) | (35.7) | (17.7) | (23.3) |
| Returned kits | N completed kits returned/total requested kits | 1/16 | 4/7 | 5/6 | 8/25 | 4/6† | 22/60 |
| | (% completing kits among total requests) | (6.3) | (57.1) | (83.3) | (32.0) | (66.7) | (36.7) |
| Test results | Negative | 1 | 3 | 4 | 8 | 4 | 20 |
| | Positive | 0 | 0 | 1 | 0 | 0 | 1 |
| | Pending‡ | 0 | 1 | 0 | 0 | 0 | 1 |
| Non-responders approached who became a responder to screening | N completed kits/N approached non-responders | 1/71 | 4/55 | 5/28 | 8/70 | 4/34 | 22/258 |
| | (% approached who became a responder) | (1.4) | (7.2) | (17.9) | (11.4) | (11.8) | (8.5) |
| Non-responders accepting the intervention who became a responder to screening | N completed kits/N accepting intervention | 1/56 | 4/46 | 5/24 | 8/61 | 4/33 | 22/220 |
| | (% accepting intervention who became a responder) | (1.8) | (8.7) | (20.8) | (13.1) | (12.1) | (10.0) |

Missing data: there were 10 missing cases for duration of intervention, five for whether intervention was accepted, two for whether leaflet was given and four for whether it was completed in the practice. The same denominator (ie, the total number of interventions carried out) applies for each question about acceptance of the intervention (ie, intervention accepted, leaflet given and leaflet completed in practice) due to issues observed in data entry. Overall four leaflets were given although intervention was ticked as not accepted and 12 leaflets were completed in practice although they were ticked as not given to the patient.
*Over 90% of the interventions (n=225) lasted up to 5 min.
†One patient from practice E requested a kit but was not sent one as s/he was only due for a new test in 2016. The National Bowel Screening System does not allow for sending additional kits for patients who are not due for another test; this helps to avoid overscreening.
‡A weak positive result (not shown) indicates that further tests are needed; in one case results for further tests were not yet available so results are shown as pending. In another case, a weak positive became a positive result after further tests.

The majority of the interventions were carried out by GPs (64.3%), followed by practice nurses (30.6%) and healthcare assistants (5.0%). Patients receiving the intervention consulted for a variety of reasons (see online supplementary file 3). The main reasons were reviews of existing conditions (22.0%), consultations due to musculoskeletal symptoms/conditions (13.2%), to carry out tests or obtain test results (11.6%) or due to respiratory or ear, nose and throat symptoms/conditions (8.8%).

The majority of patients who were offered the intervention accepted it (ie, engaged in conversation) (87.0%), with variations across practices (table 3). The leaflet was given to 74.2% of patients. Almost a quarter of the 258 patients approached (n=60) requested a replacement kit using a reply slip. Over a third of these patients (n=22) also returned a completed kit.

Younger participants accepted the intervention more often than older participants (median age 58.00; IQR 53.00–64.75 for those accepting the intervention vs 64.00; IQR 57.00–71.50 for those not accepting it). Men refused the intervention more often than women (the former represented 66.7% of all refusals). Over half (57.6%) of refused interventions were carried out by a practice nurse (see online supplementary file 3).

Descriptive data from the Bowel Screening Centre on requested kits (see online supplementary file 4) show that there was an increase in the number of requested kits across all practices during the intervention period (the highest increase in practice D and the lowest in practice E) compared with 2 years, 1 year and 6 months prior to the intervention.

### End of study evaluation: qualitative interviews

Eleven individual and one group in-depth qualitative interviews were conducted with a total of 14 primary care staff (four GPs, four practice nurses, five practice

managers and one healthcare assistant). Thirteen interviews were face to face and one was via telephone. Findings from the qualitative interviews identified four main domains: the primary care and general healthcare context; the processes involved in delivering the intervention; patient-related acceptability and primary care professional acceptability (box 1).

## Healthcare context

PCPs reported that certain organisational aspects of primary care services impacted on the implementation of the brief intervention. Existing health promotion interventions already placed demands on practices. PCPs emphasised the highly pressured primary care work environment with a cumulative impact on their ability to commit to new projects.

PCPs also highlighted important barriers to bowel screening participation such as embarrassment and practical issues, in addition to the influence of gender and ethnicity. There was variation by practitioner and also across practices, reflecting PCP's knowledge and belief in screening and also particular patient populations (with reported constraints such as illiteracy and high levels of deprivation).

## Processes in delivering the intervention

PCPs commented on the usefulness of the background information on bowel cancer and screening to increase their understanding and belief in screening and the intervention, fostering a sense of commitment. In relation to delivering the intervention itself, staff commented on appropriate types of consultations to raise the topic of screening, appropriate timing within the consultation, frequency of use of the intervention materials, adaptations made and issues around logging interventions. Staff also referred to use of computer systems to highlight non-responders (also serving as a reminder) and to log and monitor interventions, as well as the need to minimise paperwork and integrate any future interventions into existing computer systems. Constraints raised in ability to deliver the intervention related to difficulty in interacting on the topic of bowel screening with certain groups (eg, males and minority ethnic groups) but mainly to limited time in an already pressured environment.

## Patient factors and acceptability

PCPs reported that patients were positive or neutral but rarely negative when engaging on the topic of screening. They felt that patients were overall receptive to the intervention and discussing bowel screening. In a number of cases, patients had knowledge of bowel cancer and bowel screening and were aware of the benefits of taking part, but there was a large degree of perceived 'inertia', where bowel screening did not appear to be a priority.

## Primary care professional factors and acceptability

The PCPs reported the importance of increasing bowel screening participation and found the process to be acceptable, straightforward and easy to administer as part of routine consultations. However, PCPs reported

variation depending on factors such as special interests, personal experience, perceived priorities for the patient population and forced priorities as a result of limited time and work pressures. Professionals also acknowledged the influence of their attitude towards bowel screening on their approach to the intervention. Those who were motivated drew on the importance of screening and their belief in a holistic approach to healthcare, whereas for others bowel screening was not the highest priority to improve patient care.

Interviewees were cognisant of their role in the intervention process in educating patients about bowel cancer and screening and raising awareness of the benefits of participating, and how sometimes this alone was enough to prompt patients to take part. However, they felt they lacked control once the patient had left the consulting room over whether or not they ultimately returned a FOBt kit. There was also discussion of the most appropriate member of the practice team to take the intervention forward, whether this be related to time available, role (GP, practice nurse (PN) or HCA), practice load or special interest.

Practice staff reported it was feasible to roll out the intervention and made suggestions for certain adjustments to make it more effective, such as handing out kits directly to patients, streamlining any written materials and making them electronic, integrating any data recording into existing computer systems and considering funding and set time periods dedicated to bowel screening that complement other initiatives.

## End of study evaluation: questionnaires

Nineteen PCPs returned a completed questionnaire (response rate 38.8%). Thirteen were GPs, five were practice nurses and one was a practice manager. This group carried out over half (51.2%) of all interventions (n=132). As reflected in the qualitative interviews, all but one GP (no recorded interventions) stated that most patients were receptive to the intervention, that it could be easily incorporated into practice and they would theoretically be willing to take part in the study again. Nonetheless, despite positive feedback, 10 professionals highlighted lack of time as a potential or actual barrier.

Four bowel screening staff (out of seven; three screening officers and the screening supervisor) whom had been involved in the intervention returned a completed end of study questionnaire. They all stated that the intervention could be easily incorporated into their workload. However, opinion on the potential impact on workload was uncertain. Three respondents stated that it was difficult identifying calls from patients in intervention practices among over 300 daily calls. However, all four reported that it was suitable for testing in a larger study in its present form.

## Box 1 End of study interview quotes to support findings

1. Healthcare context

Existing practices

► "We remind patients about cervical screening, so it's on a, sort of, slightly similar vein." Practice nurse, practice D

► "I think similar to our alcohol brief interventions and I think it enables us to initiate conversation…about an important subject which we might not otherwise do." GP, practice D

Pressurised work environment

► "Because of the state of the practice at the minute when we've got doctors leaving—retiring and resigning—it's just put an added burden on existing people to do that." Practice manager, practice C

► "Just part of a greater workload issue. We're struggling to provide our contracted services, so I'm not going to commit to take on anything now unless it's properly resourced." GP, practice E

Acknowledging barriers to screening

► "But it's always practicalities. That's why people don't want to do it." Practice nurse, practice D

► "It's interesting because the research does show there is a kind of gradient there and that in some minority groups the uptake is not as high." GP, practice D

Knowledge of and attitude to bowel screening

► "It's definitely an extra to add in to the patients but I'm a real proponent of preventive healthcare and I think these things are worthwhile." GP, practice D

► "To be honest, the practice population that we have have far bigger problems than whether they did their bowel screening or not, so in the real world it's possibly not one of the things we would include." Practice nurse, practice A

2. Processes in implementing the intervention

Providing information on the intervention

► "Actually it was very helpful, because I hadn't understood what the patients were being asked to do." GP, practice E

► "They all thought it was a very worthwhile thing to sign up to." Practice manager, practice C

Appropriate timing and scenarios for the intervention

► "Probably if you have had any consultations that has presented with six problems and the last thing you need is to get into something else." GP, practice B

► "I think when you leave it to the end you have not encroached on the patient's time." Practice nurse, practice B

Use of intervention supporting materials

► "You maybe look at it once or twice and see what's the kind of chat and then you probably don't dig it out every time, it's so opportunistic." GP, practice A

Minimising paperwork, adaptations and integrating the intervention into existing information technology

► "Obviously if we had a reminder for everything […] then we wouldn't be able to see the screen for reminders. So it's okay in the short term but in the long term it's a bit more difficult." Practice manager, practice A

► "We've got a computer, so it tells you that you need a bowel intervention, so why (not) record the data that you wanted on the same system?" GP, practice E

Time limitations

► "I was aware that we were missing lots of people as the GPs simply didn't have time." Practice manager, practice C

► "In GP land when you've got ten minutes, ten minutes, ten minutes, then every little 5 min counts." GP, practice A

Constraints in implementing the intervention

► "It all boils down to sometimes some men don't want to discuss it. […] I've found that sometimes a barrier, especially with older men." Practice nurse, practice D

► "We have a high Asian population and they are not keen to talk about poo or the practicalities of keeping their kit beside the toilet." GP, practice D

Translating intention into action

► "When you actually spoke about it they thought it was a good idea, 'I'll do it', but whether they do it or not, don't know." Practice nurse, practice B

3. Patient factors and acceptability

Patient receptivity

► "I had no bad experiences at all. People were happy to talk about it." Practice nurse, practice E

► "I was surprised at how receptive the patients were to it." GP, practice B

Patient awareness and support for bowel screening

► "They knew pretty much what was involved." Practice nurse, practice E

► "My own finding was as soon as you mentioned it to patients and brought up screening, the majority of them were keen to go ahead and do it." GP, practice B

Patient priorities and motivation to participate in bowel screening

► "The patient would say, well that's the least of my concerns and I'll tell you why…" GP, practice A

► "Just sort of, inertia and couldn't be bothered, not a priority." GP, practice E

4. Primary care professional factors and acceptability

Acceptability to professionals

► "It isn't an onerous thing to do and what they have to do is fairly straightforward." Practice manager, practice E

Continued

## Box 1  Continued

► "It's just an extension of normal dialogues really.[…] I think it's entirely appropriate and problem, well almost problem free.[…] It was quick and simple to do and if the feedback turns out to that it's effective, then I think it would be an appropriate thing to implement in practice." GP, practice E

Professional interest, variable support and priority

► "I think there was a bit of a mixed response. I think generally GPs when they're asked to do something over and above are just like whoa, we're totally overwhelmed." GP, practice A

► "I mean they are so busy here. They are always running late […] so whether or not it just hasn't been a great priority for them." Practice nurse, practice E

Motivation to adopt the intervention

► "Bowel screening is effective and we didn't have to sell that concept to them. […] I think if they think it's a good thing they're more likely to advocate it." Practice manager, practice B

► "Sometimes it felt like quite a positive thing to do because it is about health promotion and disease prevention and that very much chimes with our ethos." GP, practice A

The intervention as part of a broader preventive health agenda

► "I think we have to be trying to educate people to look after themselves instead of fixing things after they're broken." Practice nurse, practice A

► "Giving them a message of empowering them to take control of their destiny, which is something I think that is really lacking in a population like ours." GP, practice A

The perceived professional role in educating patients and raising awareness

► "It was a good opportunity to bring it to the forefront of their consciousness […] to kind of put some medical opinion behind it and say, 'this is the reason we are doing it', you know. It does reduce your chances of having a serious bowel cancer if we catch it early." GP, practice D

► "It was just talking round the practicalities. […] 'Oh, I just didn't know how to do it' […] so you'd try to talk through it a little bit with them." Practice nurse, practice D

Potential for differing roles and involvement

► "I'm a more junior practice nurse, people aren't coming to me with loads of things, […] so maybe I have more time to look at it." Practice nurse, practice E

► "I do think that the nurses will integrate it more than the GPs will. […] I think GPs deal with the more acute problems, whereas health checks you've maybe got a bit more time and people are more relaxed and they are expecting you to ask that." Practice manager, practice A

## DISCUSSION

### Summary

Results indicate that the intervention was feasible and acceptable to PCPs and patients (as reported by professionals). Of those reached, a small but important minority became responders, a likely underestimate as email and telephone requests were not recorded. It is also possible that some patients may have completed their original kit at home and returned it. The majority of patients approached were willing to discuss the subject of bowel screening. Some patients may have made an informed choice not to participate in screening (indeed informed choice guided the intervention design), although this was not documented in this study.

Qualitative and questionnaire data indicate that the intervention was straightforward and easy to implement and reflected similar ongoing health promotion initiatives and was thus an effective way to communicate with patients about bowel screening. Overall, PCPs were willing and felt comfortable delivering the intervention in different scenarios, suggesting suitability for most primary care consultations. Practices varied in the number of patients approached and reasons for this variability were widely described in the interviews. Inappropriate or challenging scenarios reported included those involving patients with complex health and social care needs, poor literacy, English as a second language or sensitivities related to ethnicity and culture. Evidence on appropriate scenarios can help inform future interventions on how to approach these hard to reach groups. .

PCPs stated that materials were helpful to promote the intervention, draw attention to it and reinforce messages post-consultation. Nonetheless, not all practices had systems in place to identify non-responders or wished to use reminders long term; both issues can influence the success of any future implementation. Finally, feedback from the SBSC regarding the intervention was also positive, but future implementation would need to take into account the difficulty recoding telephone and email requests and explore all potential mechanisms for requests to be made.

### Study strengths and limitations

This was an evidence-based intervention informed by current data on non-participation and psychological theory, grounded in the pragmatic reality of the primary care workload. A good relationship with practices was developed. The study produced a clear audit trail and the duplicate coding of qualitative data helped ensure consistency, rigour and transparency. Nevertheless, this was a small feasibility study which requires further evaluation in larger patient populations. The study also targeted patients who consult in primary care; those who do not consult may present different challenges regarding participation. As we had to adapt to a dynamic, time-pressured primary care environment, it was not possible to record information about all eligible non-responders. It is unknown how many of them consulted (and how many of these were approached). Some elements of the intervention were

adapted by practices, this is expected in a complex intervention.[46] In fact, an intervention that can be adapted to local circumstances without loss of its essence is a strength that facilitates practical implementation.

The number of patients approached was smaller than our estimates based on national population statistics. This may be due to variation in general practice characteristics and the actual number of non-responders who consulted in primary care. Qualitative data suggest that the number of perceived inappropriate scenarios were small. Time pressures were described as the main reason for not approaching patients, even among professionals motivated to carry out the intervention. Nonetheless, we consider that an increase in requested bowel screening kits in all recruited practices was a positive outcome influenced by the intervention. More patients could be reached over time if the intervention was incorporated into practice, similar to what happens with other types of brief interventions.

Five out of the 11 invited practices completed the study. This is a low participation rate (45.5%) but not uncommon due to recognised challenges in doing research in primary care.[47 48] Reasons given for non-participation were struggles with demand/targets and pressures in primary care, a challenge also described by those who took part in the study. Only one of our recruited practices was within the 30% most deprived areas in Scotland (three were among the 50% most deprived). Hence, it is likely that these areas were under-represented (this would need to be explored in a larger study). To ensure better representation, further intervention roll-out should consider additional strategies to engage with practices in the most deprived areas (such as more personal contact and additional monetary incentives). Nonetheless, it is important to recognise that certain challenges may be beyond the scope of the study and that priorities in these practices may be different.

Finally, the context of governmental campaigns in Scotland promoting screening participation[49] and the QOF rewards may have influenced practice decisions to participate in our study. It also made it harder to separate the impact of the campaigns and the intervention on patients' behavioural response.

### Comparison with other studies

Brief interventions in primary care are well established and successful in influencing behaviours such as alcohol consumption,[50] tobacco smoking[51] and weight management.[52] Our research contributes to a body of recent UK studies examining primary care-based interventions to influence screening behaviour and demonstrating their effectiveness in improving bowel screening uptake,[13 18 19] offering further evidence on the benefits of such interventions. Our results show that intervention acceptance varied across practices with the two most deprived practices having the lowest proportion of acceptance and the lowest number of kits requested and returned. This finding suggests that GP endorsement alone is not sufficient to change patient bowel screening behaviour among the most deprived groups, as reflected in a recent study.[16]

PCPs reported lack of control after the patient leaves the consulting room—indeed only over a third of those requesting a kit actually returned it—and a gap between intention and action, a phenomenon well described in the literature.[53 54] The implementation intention plan aimed to help deal with this limitation but was seldom used by PCPs and other studies have shown mixed effectiveness.[33 35 55–57]

### Implications for practice and further research

Our feasibility parameters did not include a cost-related analysis. Future roll-out should aim to incorporate both direct costs (such as professional time) and indirect costs for the SBSC. These costs need to be balanced against long-term gains in terms of early detection and likely reduction of population mortality.[58 59]

Most interventions were carried out by a GP, which is consistent with national statistics in Scotland demonstrating that GPs carry out about two-thirds of all consultations in primary care.[60] A higher proportion of people seen by a nurse did not accept the intervention. Disease monitoring was a common reason for seeing a nurse, reflecting official data on consultation patterns in Scotland.[60] Interviews show that both GPs and PNs saw their role as important. PNs suggested that they may have more time to deliver interventions incorporated into routine patient checks and reviews, but there was some suggestion that GPs placed greater emphasis on educating and persuading patients. HCAs also reported being in a good position to deliver interventions, though the numbers in this study were small. There is scope to explore further the potential differing roles for members of the primary care team in this context and to identify ways for different professionals to have a more active involvement.

Practices varied in the number of patients approached. Our qualitative data suggest that the practice population profile, staff's level of engagement with screening, professional knowledge, experience and interests are likely to have been strong explanatory elements. Understanding this variability has implications for the flexible design of the intervention at a larger scale so it meets the needs of individual practices and different patient groups.

The primary care context was described as a highly pressured environment comprising complex patients' needs, limited financial and human resources, increasing patients but diminishing staff and the need to incentivise health promotion. When asked about the likelihood of continuing with the intervention, it was clear that despite perceiving it as useful and supporting its underlying ethos, other pressing issues would be prioritised. These challenges constrain the ability to deliver and sustain the intervention, irrespective of motivation, willingness and recognised importance. However, the flexibility of the intervention meant that it could be adapted

to suit individual practices and demonstrated an impact on bowel screening participation despite the outlined constraints.

## CONCLUSIONS

We tested a primary care-based intervention to increase uptake in non-responders to FOBt screening, and found it to be feasible and acceptable in Scottish primary practices. Nonetheless, recognised organisational and system constraints need to be considered for the intervention to be more widely implemented. Further testing in a randomised controlled trial would give robust evidence of the effectiveness of the brief intervention in increasing informed screening participation. The intervention can be useful as one tool to complement other efforts to engage with non-responders. It also reflects the broader aims from the Scottish government to raise awareness and normalise bowel screening. Our study adds to evidence that primary care can play a key role in promoting bowel screening uptake.

### Author affiliations
[1]The Usher Institute of Population Health Sciences and Informatics, Centre for Population Health Sciences, Medical School, University of Edinburgh, Edinburgh, UK
[2]Department of Psychology, Glasgow Caledonian University, Glasgow, UK
[3]Department of Psychology, University of Essex, Colchester, Essex, UK
[4]Department of Surgery and Molecular Oncology, Ninewells Hospital, University of Dundee, Dundee, UK
[5]Scottish Bowel Screening Centre, Dundee, UK
[6]University Hospitals of Coventry and Warwickshire NHS Trust, Midlands and NW Bowel Cancer Screening Hub, Rugby, UK
[7]Cancer Epidemiology Unit, University of Oxford, Oxford, UK

**Acknowledgements** Bowel screening data were obtained from the Scottish Bowel Screening Centre in Dundee; we thank all staff for their assistance and for completing the end of study questionnaires. We acknowledge the support and advice from NHS Lothian, which provided the practice lists used for recruiting practices and the help from NHS Scotland which provided the leaflets in Polish. We thank Dr Robert Scully for his help with medical acronyms/terms in the intervention proformas and Dr Ewan Gray for his advice when revising this manuscript. We also thank the reviewers for their comments and recommendations. We are very grateful to each GP, practice nurse, healthcare assistant and practice manager who agreed to take part in the study, gave up their time to be interviewed and completed the end of study questionnaires.

**Contributors** NC, DC, GV, SO, RJCS, SS, JP, DW and CC contributed to the design of the study. LB contributed to the acquisition of data for the work. NC, DC, GV, SO, RJCS, LB, SS, JP, DW and CC contributed to interpretation of data for the work. CC, DW, DC and NC were involved in recruitment, data collection and data analysis. NC, DC, GV, SO, RJCS, LB, SS, JP, DW and CC contributed to the drafting and revision of the manuscript and the approval of the final version. NC, DC, GV, SO, RJCS, LB, SS, JP, DW and CC agreed to be accountable for all aspects of the work in ensuring that questions related to the accuracy or integrity of any part of the work are appropriately investigated and resolved.

**Funding** This work was supported by Cancer Research UK (grant reference number C12357/A13965) through the National Awareness and Early Diagnosis Initiative (NAEDI-2). The funder had no role in the design, collection, analysis, and interpretation of data; in the writing of the manuscript; and in the decision to submit the manuscript for publication.

**Competing interests** None declared.

**Ethics approval** The study was approved by the South East Scotland Research Ethics Committee 01 (reference 14/SS/1067) and the NHS Lothian's Research and Development Office (Project Number 2014/0366). The Scottish Bowel Screening Governance Reference Group also approved the study.

**Provenance and peer review** Not commissioned; externally peer reviewed.

**Data sharing statement** Some of the unpublished data are available from the authors (such as intervention materials and questionnaires). The corresponding author (natalia.calanzani@ed.ac.uk) can be contacted by anyone interested in accessing these. Data from patients, primary care practices and the Scottish Bowel Screening Centre cannot be accessed by anyone who is not part of the research team due to ethical and confidentiality concerns.

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
