## [Reviewer comments · BMJ Open]

ARTICLE DETAILS

TITLE (PROVISIONAL)	Is an opportunistic primary care-based intervention for non-responders to bowel screening feasible and acceptable? A mixed methods feasibility study in Scotland
AUTHORS	Calanzani, Natalia; Cavers, Debbie; Vojt, Gabriele; Orbell, Sheina; Steele, Robert; Brownlee, Linda; Smith, Steve; Patnick, Julietta; Weller, David; Campbell, Christine

VERSION 1 - REVIEW

REVIEWER	Carlo Senore AOU Città della Salute e della Scienza SC Epidemiologia, screening, registro tumori - CPO
REVIEW RETURNED	28-Mar-2017

GENERAL COMMENTS	The authors are presenting the results of a well conducted intervention aimed to enhance the reach of an established population based screening programme, taking the opportunity of medical encounters in general practices to offer screening to subjects who did not responded to a previous screening invitation. Although very limited in size, the study was well designed and well conducted. The well structured qualitative approach allowed to derive valuable information about the barriers and opportunities for GPs and PCPs interventions. Issues related to the feasibility of the intervention would deserve, however, a more detailed analysis. The response rate of the practices was low: just 5 out of 11 candidates could be involved. This low uptake might have a negative impact on the reach of such intervention. Although the authors indicate that It is unknown how many eligible non-responders consulted (and how many of these were approached), given that they had to account for a time-pressured primary care environment, an estimate of the proportion of subjects approached in each practice might be derived, based on available data about participation rate in each practice. Considering the frequency of medical encounters in the screening age range reported in the literature, the number of people contacted over the 9-month period of the study seems low. Although this was a small pilot study, focused on the feasibility of the intervention, some estimates of the costs of the intervention (inkling the time devoted by the primary care professionals) might be useful to have an idea of the expected burden of the intervention and of the cost per subject attending screening. The authors should also explain why most of the interventions were
---

	actually delivered by the GPs, while in the design of the study they were apparently planned as a task of PC professionals, like practice nurses.
--	---

REVIEWER	Greg Rubin Durham University UK
REVIEW RETURNED	24-Apr-2017

GENERAL COMMENTS	A feasibility study of a primary care intervention to increase uptake of bowel cancer screening. Methods: The intervention is carefully developed. There are a number of theoretical models on which such interventions are based; I was unclear why the ones specified were chosen, rather than more widely used models. Some justification is merited. Results: These are well presented. What is lacking, given this is a feasibility study, is information about the number of people who consulted but did not receive the intervention. From tables 1 and 2 I estimate a 4 fourfold difference in the proportion of people approached who had not participated in screening. Greater characterisation of those patients who didn't get the intervention would be important. Secondly, no attempt was made to directly assess acceptability to patients. This is a significant problem that would impact on the feasibility of any future trial. These points should be addressed in the discussion, if nothing else. Discussions: does not speculate beyond the findings and the specifically exploratory nature of the study. Otherwise, this is a well-written paper that addresses an issue of interest to primary care physicians and those responsible for screening programmes
--

VERSION 1 – AUTHOR RESPONSE

Reviewer: 1

Reviewer Name: Carlo Senore

Please leave your comments for the authors below

1. The authors are presenting the results of a well conducted intervention aimed to enhance the reach of an established population based screening programme, taking the opportunity of medical encounters in general practices to offer screening to subjects who did not responded to a previous screening invitation. Although very limited in size, the study was well designed and well conducted. The well structured qualitative approach allowed to derive valuable information about the barriers and opportunities for GPs and PCPs interventions.

Thank you for your comments.

2. Issues related to the feasibility of the intervention would deserve, however, a more detailed analysis. The response rate of the practices was low: just 5 out of 11 candidates could be involved. This low uptake might have a negative impact on the reach of such intervention.

Thank you; we agree that we need to explore the potential impact of non-participation further and acknowledge that it is possible we may have missed important groups.

There are recognised challenges in doing research in primary care. Current pressures faced by these services in the UK make it a difficult environment for care provision and even more for research (1, 2). We considered these challenges when designing the intervention.

We think it is positive that, despite recognised constraints, over half of the invited practices agreed to take part and five of them completed the study. Participants were enthusiastic about the intervention and considered it to be feasible and acceptable. Results indicate that acceptance was helped by the fact that professionals considered the intervention to be important. Even if we estimate a 45% RR rate from practices in a large scale study, results could be substantial in terms of increasing bowel screening participation.

Evidence shows that, in Scotland, primary care practices in more deprived areas have fewer GPs than those in the least deprived regions (3), and that these practices are struggling to deal with complex, increasing demands from their population (4). We have checked the practices included in Wave 1 of Recruitment and we can see that the practices refusing to take part were in more deprived regions (decile ranges 2.1-4.6 compared to decile ranges 2.6-6.2 in the recruited group), with larger practice list sizes (from 8,287 to 13,984 registered patients), and consequently larger average number of non-responders. We have now added a table with this information to Supplementary File 2. Although this was a small feasibility study and this trend would need to be checked in a larger study, results indicate that more deprived practices could be underrepresented when rolling out the intervention. As non-participation in Scotland is higher amongst the most deprived (5), we could miss opportunities to engage with patients in this important group (provided that they consulted in primary care).

We have now acknowledged these possibilities in the Strengths and Limitations Section (pages 17-18) in the main manuscript. We are also proposing ways to tackle the problem (while also recognising that some issues are beyond the scope of the study). Please note that page numbers refer to the document with tracked changes (as opposed to the cleaned file):

“Five out of the 11 invited practices completed the study. This is a low participation rate (45.5%), but not uncommon due to recognised challenges in doing research in primary care [47, 48]. Reasons given for non-participation were struggles with demand/targets and pressures in primary care, a challenge also described by those who took part in the study. Only one of our recruited practices was within the 30% most deprived areas in Scotland (three were amongst the 50% most deprived). Hence, it is likely that these areas were underrepresented (this would need to be explored in a larger study). To ensure better representation, further intervention roll-out should consider additional strategies to engage with practices in the most deprived areas (such as more personal contact and additional monetary incentives). Nonetheless, it is important to recognise that certain challenges may be beyond the scope of the study, and that priorities in these practices may be different.”

References:

1. Baird B, Charles A, Honeyman M, et al. Understanding pressures in general practice. The King's Fund; 2016. p. 1-100.
2. Wilson S, Delaney BC, Roalfe A, et al. Randomised controlled trials in primary care: case study. *BMJ*. 2000;321(7252):24.
3. Blane DN, McLean G, Watt G. Distribution of GPs in Scotland by age, gender and deprivation. *Scottish Medical Journal*. 2015;60(4):214-9.
4. Watt G, Brown G, Budd J, et al. General Practitioners at the Deep End: The experience and views of general practitioners working in the most severely deprived areas of Scotland. Occasional paper (Royal College of General Practitioners). 2012(89):i-40.
5. Scottish Bowel Screening Programme. Scottish Bowel Screening Programme. Key Performance Indicators Report: November 2015 data submission. Invitations between 1st May 2013 and 30th April 2015: Available from: <https://www.isdscotland.org/Health-Topics/Cancer/Publications/2016-02-23/2016-02-23-Bowel-Screening-KPI-Report.xlsx>. Accessed 25 May 2017.

3. Although the authors indicate that it is unknown how many eligible non-responders consulted (and how many of these were approached), given that they had to account for a time-pressured primary care environment, an estimate of the proportion of subjects approached in each practice might be derived, based on available data about participation rate in each practice. Considering the frequency of medical encounters in the screening age range reported in the literature, the number of people contacted over the 9-month period of the study seems low.

We understand the reviewer's concerns about the number of people contacted during the nine-month period (bearing in mind that each practice carried out the intervention for four months). When developing the study we had estimated the study population based on available population statistics (Supplementary file 1). In order to commence practice recruitment we later received data on % uptake, population aged 50-75 and average monthly non-responder numbers for every general practice in Lothian. We have now calculated our estimates of the proportion of subjects approached in each practice taking into account the characteristics of our included practices. These are rough estimates due to limited data available. Furthermore, they are still based on the assumption that 86% of patients consult in primary care at least one a year (Supplementary File 1), and that non-responders are well-represented among this group. Finally, to facilitate calculations we are assuming that the number of consultations are similar in every month of the year. We had prepared a table with these calculations, but the formatting was lost when submitting our reply online. Therefore, we are describing our calculations in text instead. We trust that this is acceptable.

In Practice A, 1,215 patients (86% out of 1,413 patients aged 50-75 – described in Table 1 in the manuscript) would have consulted in a year, and a third of them (n=405) would have consulted during the 4-month study period. Based on the reported uptake of <45% in this practice, we estimated that 55% of these patients could be non-responders, resulting in 223 potentially eligible patients during the study period. Our study findings reported that 71 patients were approached during the study in this practice.

Using the same rationale and data provided in Table 1, Practice B would have a maximum number of 419 potentially eligible patients consulting during the study period (55 were approached in the study); numbers are 361 for Practice C (28 patients were approached in the study), 178 for Practice D (70 patients were approached in the study) and 215 for Practice E (34 patients were approached in the study).

Based on these rough estimates, the proportion of non-responders approached among potentially eligible patients varied widely across practices (from less than one in 10 to almost four in 10 patients). We do not have sufficient statistical information to assess reasons for this variation (although results from qualitative interviews help to shed light on reasons why patients were not approached). It is important to highlight that this is an opportunistic intervention. Hence, the decision to approach a patient was dependent on the professionals' priorities, motivation, whether they remembered to do it, their views on the appropriateness of approaching the topic with a patient or having sufficient time to carry out the intervention. This is reflected in the actual versus the potential number of people whom could be approached. Nonetheless, if the intervention became common practice it is likely that a higher number of patients would be approached over time. We have now added a paragraph to the discussion section acknowledging that the number of patients approached was lower than what we had estimated when designing the study (Section Strengths and Limitations, page 17). As previously described in the conclusion, we believe that the intervention can help to make a difference as one approach alongside other ongoing public health initiatives to promote screening uptake:

“The number of patients approached was smaller than our estimates based on national population statistics. This may be due to variation in general practice characteristics and the actual number of non-responders who consulted in primary care. We may have underestimated the number of possible appropriate consultations, although qualitative data suggest that the number of perceived inappropriate scenarios were small. Furthermore, time pressures were described as the main reason for not approaching patients, even among professionals motivated to carry out the intervention. Nonetheless, we consider that an increase in requested bowel screening kits in all recruited practices was a positive outcome influenced by the intervention, and that more patients could be reached over time if the intervention was incorporated into practice, similar to what happens with other types of brief interventions.”

4. Although this was a small pilot study, focused on the feasibility of the intervention, some estimates of the costs of the intervention (including the time devoted by the primary care professionals) might be useful to have an idea of the expected burden of the intervention and of the cost per subject attending screening.

This was a small study testing an opportunistic intervention to be used during routine 10-minute

consultations, and we have not incorporated a cost-analysis. In theory, the costs would be minimal as the intervention becomes part of usual work (especially considering that practices are incentivised by the government to develop strategies to engage with non-responders). However, we acknowledge that the estimated three minutes used for the intervention could have been used to address other issues which may have cost implications for the national health services.

If we were to calculate the direct costs of the intervention (i.e. the average two minutes used during a consultation) we would use the Personal Social Services Research Unit (PSSRU) unit costs 2016 (6), which list £3.90 per minute for a GP consultation or £7.80 for a two-minute intervention. Costs for a consultation with a nurse would vary (range of £22-£73 per hour, or £0.73-2.43 for a 2-minute intervention). The mean annual basic pay for a health care assistant is £17,012 (equivalent to an £8.18 hourly rate or £0.27 for a two-minute intervention if the professional was working 8 hours a day, 5 days a week).

There would also be costs to print the leaflet for patients and prepare freepost envelopes (1000 envelopes cost £100.77 while 1,000 leaflets cost £343.71 - £444.48 overall which was paid by the research team). Postage costs for returning a reply slip are now £0.65 per envelope (equivalent of £650 for 1,000 envelopes); these were paid by the Scottish Bowel Screening Centre. Importantly, patients could have contacted them by email or telephone instead. The telephone is free to the user but may represent marginal additional costs for the Screening Centre.

For information, we have estimated the costs of the intervention for general practices (including the intervention supporting materials). Again, due to formatting restrictions we are describing our calculations in text instead of using a table.

- In Practice A, 33 interventions were carried out by a GP (33*£7.8), and 38 by a practice nurse (38*£1.58 – midrange costs). Costs for leaflet and envelopes would have been £31.56 for the 71 patients. Total estimated costs would be £349.00.
- In Practice B, 31 interventions were carried out by a GP (31*£7.8), 11 by a practice nurse (11*£1.58) and 13 by a Health Care Assistant (13*£0.27). Costs for leaflet and envelopes would have been £24.45 for the 55 patients. Total estimated costs would be £287.14.
- In Practice C, 21 interventions were carried by a GP (21*£7.8), and 7 by a practice nurse (7*£1.58 – midrange costs). Costs for leaflet and envelopes would have been £12.45 for the 28 patients. Total estimated costs would be £187.31.
- In Practice D, 50 interventions were carried by a GP (50*£7.8), and 20 by a practice nurse (20*£1.58 – midrange costs). Costs for leaflet and envelopes would have been £31.11 for the 70 patients. Total estimated costs would be £452.71.
- In Practice E, 31 interventions were carried by a GP (31*£7.8), and 3 by a practice nurse (3*£1.58 – midrange costs). Costs for leaflet and envelopes would have been £15.11 for the 34 patients. Total estimated costs would be £261.35.

Overall, 166 interventions were carried out by GPs (166*£7.8), 79 by practice nurses (79*£1.58) and 13 by Health Care Assistants (13*£0.27). Costs for leaflets and envelopes would have been £114.68. Total estimated costs for the intervention would be £1,537.81.

Based on these estimates, the average direct costs for a 4-month intervention by practice would be £307.56 (i.e. £1,537.81 divided by five), but as calculations show these costs would vary widely depending on how many patients were approached and which professionals were carrying out the intervention. As a cost-analysis was not planned when developing the study and it is not part of our predefined feasibility parameters, we would prefer not to include this analysis a posteriori. We trust this is acceptable.

Calculating indirect costs would be more complex. Although these analyses are important, we believe they are beyond the scope of this small feasibility study to estimate. Published economic evaluations (7) indicate a cost of £11.74 for each completed FOBt test, and describe marginal costs for different

screening strategies. Although an increase in screening uptake may have cost implications for the bowel screening centre, this is likely to be offset by the gains of early detection. Costs do increase with increased uptake, but so does the effectiveness in terms of reducing population mortality (8). Furthermore, as treatment costs decrease substantially when cancer is detected at an earlier stage (which is a benefit of bowel screening) (9), we believe that the intervention would not be a burden to the health services (instead, it would be beneficial). We have consulted with a Health Economist, who advised us that in order to obtain the most methodologically correct estimates we would need to reanalyse a model such as the one presented in the reference above (7), considering uptake levels with and without the intervention. We believe this should be something to consider in a larger, different study.

We have added a sentence to the manuscript (Implications for Practice and Further Research subsection, Page 19), acknowledging that we did not carry-out a cost analysis, and that if the study was to be rolled out in a large-scale trial this should be incorporated:

“Our feasibility parameters did not include a cost-related analysis. Future roll-out should aim to incorporate both direct costs (such as professional time) and indirect costs for the SBSC. These costs need to be balanced against long-term gains in terms of early detection and likely reduction of population mortality [58, 59].”

References

6. Curtis L, Burns A. Unit Costs of Health and Social Care 2016: Personal Social Services Research Unit, University of Kent, Canterbury; 2016.
7. Tappenden P, Chilcott J, Eggington S, et al. Option appraisal of population-based colorectal cancer screening programmes in England. *Gut*. 2007;56(5):677-84.
8. Sharp L, Tilson L, Whyte S, et al. Cost-effectiveness of population-based screening for colorectal cancer: a comparison of guaiac-based faecal occult blood testing, faecal immunochemical testing and flexible sigmoidoscopy. *Br J Cancer*. 2012;106(5):805-16.
9. Birtwistle M, Earnshaw A. Saving lives, averting costs. An analysis of the financial implications of achieving earlier diagnosis of colorectal, lung and ovarian cancer Incisive Health on behalf of Cancer Research UK; 2014. Available from: http://www.cancerresearchuk.org/sites/default/files/saving_lives_averting_costs.pdf.

5. The authors should also explain why most of the interventions were actually delivered by the GPs, while in the design of the study they were apparently planned as a task of PC professionals, like practice nurses.

We had estimated that other professionals (especially practice nurses) would be more suited for carrying out the intervention, especially in routine consultations to deal with other chronic conditions such as hypertension and diabetes. During the information sessions we also emphasised that the intervention was suitable for all health professionals, and this was also highlighted in the supporting materials. Interviews show that nurses believed they had more time and their types of consultations were appropriate for incorporating the intervention, but they also describe how GPs perceived that having conversations about screening was part of their usual work and an important part of their role.

A possible explanation for our results is that most consultations in primary care are still carried out by GPs. The latest available information on consultations by health care professionals in Scotland states that two-thirds of consultations in primary care are carried out by them, with the remaining third being carried out by nurses and other professionals (10). With this information in mind, the proportion of interventions carried out by practice nurses and other professionals was actually higher than expected in Practices A and B, slightly lower than expected in Practice C and Practice D, and much lower than expected in Practice E (Table 2 in the manuscript). We do believe that there are still underlying issues that could be further investigated in a larger study. We have now amended a paragraph in the Implications for Practice and Further Research subsection (Page 19):

“Most interventions were carried out by a GP, which is consistent with national statistics in Scotland demonstrating that GPs carry out about two-thirds of all consultations in primary care [60]. A higher proportion of people seen by a nurse did not accept the intervention. Disease monitoring was a common reason for seeing a nurse, reflecting official data on consultation patterns in Scotland [60].

Interviews show that both GPs and PNs saw their role as important. PNs suggested that they may have more time to deliver interventions incorporated into routine patient checks and reviews, but there was some suggestion that GPs placed greater emphasis on educating and persuading patients. HCAs also reported being in a good position to deliver interventions, though the numbers in this study were small. There is scope to explore further the potential differing roles for members of the primary care team in this context, and to identify ways for different professionals to have a more active involvement.”

References

10. ISD Scotland. Practice Team Information (PTI) Annual Update (2012/13). Available from: <http://www.isdscotland.org/Health-Topics/General-Practice/Publications/2013-10-29/2013-10-29-PTI-Report.pdf>. Accessed 25 May 2017.

Reviewer: 2

Reviewer Name: Greg Rubin

Please leave your comments for the authors below

1. A feasibility study of a primary care intervention to increase uptake of bowel cancer screening. Methods: The intervention is carefully developed. There are a number of theoretical models on which such interventions are based; I was unclear why the ones specified were chosen, rather than more widely used models. Some justification is merited.

Thank you for your comments. Due to limitation in word count and in order to be consistent in terms of our messages, we have developed two manuscripts for the study. The first one (currently under review by a different journal) focuses on the process of developing the intervention, in which there is a thorough justification of our choices of theoretical frameworks. We acknowledge that by having two manuscripts, relevant information on intervention development is not available in this document.

The Health Behaviour Framework is a synthesis of major health behaviour models including the Social Cognitive Theory, the Health Belief Model, the Theory of Planned Behaviour, the Transtheoretical model and Social Influence Theory. We chose this framework as it considers the context in which interventions are introduced, including characteristics of the service provider and the health care setting as well as larger community and societal influences. In this way, the health behaviour framework acknowledges that behavioural changes do not take place in isolation. The transfer of intent to actual behaviour depends on the interplay between a number of perceived barriers and the presence of supporting factors in the individual, across health care and/or society. This framework has also been successfully applied to cancer screening health behaviours (11-13).

Furthermore, we have incorporated Implementation Intentions as they have been shown to be associated with higher participation in studies about cervical (14), breast (15) and bowel (16) screening, although they did not seem to have a positive impact on other studies about bowel (17) and cervical (18) screening. We wished to see whether its adoption would have any impact in our study.

Motivational interviewing has been shown to increase bowel screening participation (19, 20), and its principles aided the development of open, non-directive statements to introduce the topic of non-participation.

We have now added brief information to the Materials and Methods section, Intervention subsection (Page 6), and we trust that this addresses the reviewers' concerns. Please note that page numbers refer to the document with tracked changes (as opposed to the cleaned file):

“We chose the Health Behaviour Framework as it synthesises major health behaviour models while also considering contextual factors. It has also been successfully applied to cancer screening health behaviours [30-32]. Implementation Intentions have been associated with higher participation in studies about cervical [33], breast [34] and bowel [35] screening. Finally, motivational interviewing has been shown to increase bowel screening uptake [36, 37]; its principles aided the development of non-directive statements to discuss non-participation.”

References

11. Tu SP, Yip MP, Chun A, et al. Development of intervention materials for individuals with limited English proficiency: lessons learned from "Colorectal Cancer Screening in Chinese Americans". *Medical care*. 2008;46(9 Suppl 1):S51-61.
12. Jo AM, Maxwell AE, Wong WK, et al. Colorectal cancer screening among underserved Korean Americans in Los Angeles County. *Journal Of Immigrant And Minority Health / Center For Minority Public Health*. 2008;10(2):119-26.
13. Maxwell AE, Bastani R, Crespi CM, et al. Behavioral mediators of colorectal cancer screening in a randomized controlled intervention trial. *Preventive medicine*. 2011;52(2):167-73.
14. Sheeran P, Orbell S. Using implementation intentions to increase attendance for cervical cancer screening. *Health Psychol*. 2000;19(3):283-9.
15. Rutter D, Quine L, Steadman L, et al. Increasing Attendance at Breast Cancer Screening: Field Trial. Final Report to NHSBSP: University of Kent. Department of Psychology. Centre for Research in Health Behaviour; 2007.
16. Greiner KA, Daley CM, Epp A, et al. Implementation Intentions and Colorectal Screening: A Randomized Trial in Safety-Net Clinics. *Am J Prev Med*. 2014;47(6):703-14.
17. Lo SH, Good A, Sheeran P, et al. Preformulated implementation intentions to promote colorectal cancer screening: a cluster-randomized trial. *Health Psychol*. 2014;33(9):998-1002.
18. Walsh JC. Increasing Screening Uptake for a Cervical Smear Test: Predictors of Attendance and the use of Action Plans in Prior Non-Attendees. *The Irish Journal of Psychology*. 2005;26(1-2):65-73.
19. Ling BS, Schoen RE, Trauth JM, et al. Physicians encouraging colorectal screening: a randomized controlled trial of enhanced office and patient management on compliance with colorectal cancer screening. *Archives of internal medicine* 2009; 169:47-55.
20. Lasser KE, Murillo J, Medlin E, et al. A multilevel intervention to promote colorectal cancer screening among community health center patients: results of a pilot study. *BMC Family Practice* 2009; 10:1-7.

2. Results: These are well presented. What is lacking, given this is a feasibility study, is information about the number of people who consulted but did not receive the intervention. From tables 1 and 2 I estimate a 4 fourfold difference in the proportion of people approached who had not participated in screening. Greater characterisation of those patients who didn't get the intervention would be important.

Unfortunately we do not have any quantitative information on non-responders who were not approached about the intervention. However, professionals' views on inappropriate scenarios shed light on some of their characteristics (patients with complex health and social care needs, poor literacy, English as a second language, or sensitivities related to ethnicity and culture). This is described in the Summary (as part of the Discussion section, pages 16-17).

Reviewer 1 (Question 3) has also asked a similar question, and we have attempted to calculate the proportion of patients approached amongst potentially eligible non-responders. Based on our estimates, the proportion of non-responders approached varied widely across practices (from less than one in 10 to almost four in 10 potentially eligible patients). We have now acknowledged in the Strengths and Limitations Section (page 17) that we may have missed non-responders while carrying out the intervention:

"The number of patients approached was smaller than our estimates based on national population statistics. This may be due to variation in general practice characteristics and the actual number of non-responders who consulted in primary care. We may have underestimated the number of possible appropriate consultations, although qualitative data suggest that the number of perceived inappropriate scenarios were small. Furthermore, time pressures were described as the main reason for not approaching patients, even among professionals motivated to carry out the intervention. Nonetheless, we consider that an increase in requested bowel screening kits in all recruited practices was a positive outcome influenced by the intervention, and that more patients could be reached over time if the intervention was incorporated into practice, similar to what happens with other types of brief interventions."

3. Secondly, no attempt was made to directly assess acceptability to patients. This is a significant

problem that would impact on the feasibility of any future trial. These points should be addressed in the discussion, if nothing else.

We agree that the study could have benefited from directly contacting patients who took part in the intervention. We did not have any direct access to patient information (only anonymised data provided in intervention proformas). We did, however, assess the acceptability of the intervention with eight health care professionals and 19 purposively selected non-responders to bowel screening when developing the intervention (this is described in the other manuscript about the study - currently under review). The study was approved by the West of Scotland REC Committee 1 reference 13-WS-0006. Both professionals and non-responders were supportive of the brief intervention. The latter specified that they expected the health care professional to take a broader approach towards their health, and that they would not mind having the topic of screening raised during a routine consultation for other purposes. These interviews allowed for the identification of possible bottlenecks during implementation and resulted in some changes to the intervention design (such as adding an email address to the leaflet as an additional way to request a test kit).

As these data are yet to be published, we have opted for adding further information to the methods section instead (Materials and Methods section, Intervention subsection, Page 6); we trust that this is acceptable:

"The process of developing the intervention also included eight interviews with health professionals and 19 non-responders to bowel screening to explore their views on its acceptability in a primary care setting. Overall feedback was positive and results are reported elsewhere (manuscript under review)."

4. Discussions: does not speculate beyond the findings and the specifically exploratory nature of the study. Otherwise, this is a well-written paper that addresses an issue of interest to primary care physicians and those responsible for screening programmes

Thank you for your comments.

In previous manuscript drafts we had tried to speculate beyond some of our findings, although we opted for removing these comments (mainly due to word count, but also to have a more concise discussion section). Previously we had discussed possible reasons why our intervention reached younger groups more often (older people are less represented among non-responders, it may be a reflection of younger practice populations, or perhaps professionals considered inappropriate to approach older people due to high levels of multi-morbidity in this group). We had also discussed how patients may have made an informed decision not to take part in screening, and how important it is to consider the possibility of coercion in the context of our intervention (something that was carefully reflected upon when designing the study). Finally, we had also further explored the variation of patients approached by practices. Our qualitative data had suggested that the practice population profile, staff's level of engagement with screening, professional knowledge, experience and interests are likely to have been strong explanatory elements. These characteristics also help to explain variations when implementing the intervention (such as creating a digital proforma, doing interventions over the telephone, creating systems to avoid engaging with the same patient twice and completing the leaflet with the patient). We believe that understanding this variability has implications for the flexible design of the intervention at a larger scale so it meets the needs of individual practices and different patient groups.

We have now opted for including the latter issue in the manuscript, as part of the Implications for Practice and Further Research subsection (Page 19):

"Practices varied in the number of patients approached. Our qualitative data suggest that the practice population profile, staff's level of engagement with screening, professional knowledge, experience and interests are likely to have been strong explanatory elements. Understanding this variability has implications for the flexible design of the intervention at a larger scale so it meets the needs of individual practices and different patient groups."